# Antihypertensive Treatment in Diabetic Kidney Disease: The Need for a Patient-Centered Approach

**DOI:** 10.3390/medicina55070382

**Published:** 2019-07-16

**Authors:** Barbara Bonino, Giovanna Leoncini, Salvatore De Cosmo, Eulalia Greco, Giuseppina T. Russo, Annalisa Giandalia, Francesca Viazzi, Roberto Pontremoli

**Affiliations:** 1Department of Internal Medicine, University of Genoa and Ospedale Policlinico San Martino, Viale Benedetto XV 6, 16132 Genoa, Italy; 2IRCCS Casa Sollievo della Sofferenza, Viale Cappuccini 1, 71013 San Giovanni Rotondo (FG), Italy; 3Department of Clinical and Experimental Medicine, University of Messina, Piazza Pugliatti 1, 98122 Messina, Italy

**Keywords:** diabetes, blood pressure target, RAAS inhibitors, SGLT-2 inhibitors, renal outcome, cardiovascular outcome

## Abstract

Diabetic kidney disease affects up to forty percent of patients with diabetes during their lifespan. Prevention and treatment of diabetic kidney disease is currently based on optimal glucose and blood pressure control. Renin–angiotensin aldosterone inhibitors are considered the mainstay treatment for hypertension in diabetic patients, especially in the presence of albuminuria. Whether strict blood pressure reduction entails a favorable renal outcome also in non-albuminuric patients is at present unclear. Results of several clinical trials suggest that an overly aggressive blood pressure reduction, especially in the context of profound pharmacologic inhibition of the renin–angiotensin–aldosterone system may result in a paradoxical worsening of renal function. On the basis of this evidence, it is proposed that blood pressure reduction should be tailored in each individual patient according to renal phenotype.

## 1. Introduction

Diabetic kidney disease (DKD) is a relatively common and feared complication of both type 1 and 2 diabetes. It affects 30% to 40% of diabetic patients over their lifespan and negatively impacts their cardiovascular risk burden and overall survival [1]. Retarding the progression of overt DKD to end-stage renal disease (ESRD) has proved a challenging task and prevention of its onset may well be a more cost-effective strategy to reduce the burden of risk for these patients. Among established therapeutic interventions, beside optimal glucose control [2,3], antihypertensive treatment by means of renin–angiotensin–aldosterone system inhibitors (RAAS-is) has traditionally been considered the treatment of choice [4]. However, the effectiveness of this intervention in real-life clinical conditions has probably been overemphasized, and optimal blood pressure (BP) values in order to maximize the treatment benefit are still a matter of debate [5].

A recently published retrospective study while confirming that long-term, strict control of BP is associated to a reduction of albuminuria suggested that the related benefit in terms of glomerular filtration rate (GFR) preservation may be smaller than anticipated because of a paradoxical worsening in renal function at very low BP levels [6]. As a matter of fact, several other studies seem to confirm these data, indicating a lack of renal benefit when very low BP values are achieved [7,8]. This could probably be related to an impaired adaptation of intra-renal hemodynamics to BP variations because of vascular stiffening and atherosclerosis.

Experimental and clinical studies have convincingly demonstrated the involvement of increased activity of the RAAS both at the systemic and tissue level in the development and progression of diabetic renal damage [9]. Thus, RAAS-is are considered the mainstay treatment of hypertension in diabetic patients, particularly in those patients with overt proteinuria [4]. angiotensin-converting-enzyme inhibitor (ACE-is) and angiotensin II receptor blockers (ARBs) have been shown to effectively improve intra-renal hemodynamics, thereby conveying long-term renal protection. Moreover, RAAS-is are known to attenuate interstitial fibrosis and glomerular sclerosis at the renal tissue level via multiple mechanisms, including a reduction of angiotensin II action as well as a reduction in the production of platelet-derived growth factor and connective tissue growth factors [10].

Albuminuria is a known marker of kidney damage with an established prognostic role both in cardiovascular and in renal risk. Recent, long-term studies confirm that a decrease in albuminuria obtained with optimal BP control is associated with better cardiovascular and renal outcome, while an increase in urine albumin excretion over time entails worse prognosis [11,12]. Thus, one might state that changes in albuminuria translate in parallel changes of risk and albuminuria reduction could be taken as a treatment target in itself [13]. However, recent studies have shown that both in Type 2 and Type 1 diabetes, CKD is a rather heterogeneous phenotype, with different underlying histologic patterns, with fibrosis and ischemia variably involving glomeruli and the interstitium [14]. Indeed, recent surveys seem to indicate that albuminuria is absent or minimal in a considerable proportion of diabetic patients presenting with declining renal function [15].

While lower BP values and maximal pharmacologic inhibition of the RAAS have consistently been shown to improve renal outcome in diabetic patients presenting with the traditional “albuminuric” renal phenotype, it is uncertain whether a similar therapeutic strategy should be applied in the absence of albuminuria. Thus, on the basis of more recent epidemiological data, it has been proposed that antihypertensive treatment should be individualized both in terms of target BP values and possibly, in terms of antihypertensive drugs combination according to different presenting renal phenotypes [5].

## 2. Blood Pressure Reduction and Renal Protection: Which is the Target Value?

GFR and albuminuria are usually considered to be valid biomarkers that can be used to monitor renal function and to assess the effectiveness of therapeutic strategies for renal protection [16]. The results of many clinical trials have shown that changes in albuminuria over time are related to cardiovascular and renal protection, emphasizing the usefulness of achieving optimal BP targets in clinical practice [12]. Two recent meta-analyses [11,17] seem to confirm a benefit in terms of cardiovascular and renal risk when a reduction of albuminuria is obtained, independent of BP reduction and the type of antihypertensive treatment. On the basis of these results, reduction in albuminuria has been proposed as an independent target of treatment [18]. However, there have also been conflicting views about the significance of albuminuria changes in relation to renal outcome. In fact, albuminuria and GFR modifications may follow different paths in time under the effect of antihypertensive treatment. Some studies suggest that reduction of albuminuria may in part be the result of GFR reduction itself and caution should be taken when interpreting data, especially in the short-term [6]. Recently, a large trial has compared two different therapeutic regimens aimed at lowering BP either to below 140 or below 120 mmHg in a large group of patients with high cardiovascular risk [19]. More intensive BP reduction was generally associated to better cardiovascular outcome and lower mortality, although no benefit on renal endpoints was evident, and some renal function worsening was even present in specific subgroups. These results, however, seem to be difficult to translate into clinical practice due to severe methodological limitations in the study, especially regarding the technique of BP recording [20]. Nonetheless, results of the SPRINT study conducted in a non-diabetic cohort seem to be in line with those from the ACCORD study, carried out in patients with Type 2 diabetes. In fact, the latter showed that intensive BP control is not associated to cardiovascular and renal protection, with the only exception being the risk of stroke, which keeps decreasing at very low BP levels. Furthermore, two meta-analyses have shown that low BP is associated with better cardiovascular protection in patients at high risk, including those with diabetes [7,8]. However, cardiovascular benefits seem to fade at BP values below 130 mmHg and lower values (i.e., below 120 mmHg) may even entail more risk than benefit, with the only exception of further reduction in cerebrovascular events. Based on this evidence, International Guidelines have recently revised recommendations on optimal BP values indicating that high-risk patients, such as those with CKD and/or diabetes, should be treated to achieve systolic BP values <130 mmHg. Interestingly though, current European Guidelines [4] also recommend, for the first time, a threshold below which BP should not be lowered further in these patients to prevent a paradoxical increase in morbidity and mortality. Considering the positive correlation between the degree of albuminuria and risk of renal progression, patients with proteinuria should be maintained at a BP value below 130 mmHg and possibly even lower [21]. In the absence of albuminuria, a BP target below 130 mmHg but above 120 mmHg appears to be wise [22]. Real-life data seem to confirm that a J-curve describes the relationship between BP reduction and renal morbidity. In the Italian Associazione Medici Diabetologi (AMD) database, GFR was worse in the subgroup of patients with a systolic BP under 130 mmHg [23].

Many diabetic hypertensive patients may suffer from autonomic neuropathy, and therefore, from orthostatic hypotension and other features of impaired systemic hemodynamics. Thus, in clinical practice, seated or standing BP should be used as a target for treatment and great care should be paid when titrating antihypertensive drugs, especially peripheral alfa-blocking drugs. In addition, the timing of drug administration should be tailored in each individual patient and attention should be paid to avoid the risk of haemodynamic changes in patients in the presence of diarrhea or hypovolemia.

## 3. Glomerular Haemodynamics as a Target for Treatment: Have We Reached the Limit?

RAAS inhibitors are considered the mainstay treatment for a variety of cardiovascular conditions such as of hypertension, diabetes, proteinuria, and heart failure and myocardial infarction because of their beneficial effects on cardiovascular protection [4]. In particular, the effect on renal protection is related to anti-fibrotic properties and the reduction of proteinuria. Start and up-titration of RAAS-is are often the causes of acute worsening in renal function which could have been monitored [24]. A real-life study, conducted in 2017 on more than 100,000 patients with different comorbidities in the UK, investigated the relationship between worsening renal function and cardiovascular and renal outcome after initiation of RAAS-is. This study showed that the decline of GFR was related to adverse cardio renal outcomes [25]. Additional doubts on the renal safety of multiple pharmacological interventions have been raised in recent years. Different trials such as ONTARGET, ALTITUDE and NEPHRON D showed in a variety of high-risk patients that profound blockade of RAAS results in greater loss of renal function and higher risk of hyper-kalemia with no advantages in short- or long-term renal outcomes [26,27,28].

Recently, sodium/glucose cotransporter 2 inhibitors (SGLT2-is) have received attention as a class of promising hypoglycemic agents with additional ability to slow down the progression of diabetic kidney disease. They improve glucose metabolism by inhibiting glucose reabsorption at the proximal part of the renal tubule, thereby increasing glycosuria. Moreover, SGLT2-is have been shown to improve the systemic hemodynamic profile by reducing BP and promoting weight loss. While renal changes observed with the use of these agents may be due to multiple pathogenic mechanisms, a large part of their renal protective effect seems to be mediated by their ability to favorably interfere with renal hemodynamics. In fact, SGLT2-is attenuate hyperfiltration and lower intra-glomerular pressure by increasing afferent arteriolar vaso-constriction, as a consequence of tubular glomerular feed-back activation at the macula densa [29]. Besides these hemodynamic effects, renal protection is increased by a number of favorable metabolic effects. In fact, weight loss and a negative caloric balance are fostered by urinary loss of glucose. Increased glucagone production and reduced insulin release may promote gluconeogenesis and lipolysis and translate into a reduction of fat tissue. In addition, the observed uricosuric effect due to decreased GLUT-9-mediated urate reabsorption at the tubular level may contribute to an improvement in the cardiorenal metabolic profile. Ischemic renal damage may further be prevented through other still uncertain mechanisms such as the increase of hypoxia-inducible factor production and reduction in vascular stiffness [30]. The reno-protective properties of glycosuric agents is additional to that of traditional RAAS-is and appears to complement it.

It has been estimated that by using SGLT2-is in addition to RAAS-is, an additional 30%–40% renal protection benefit could be obtained in clinical practice. While this approach looks very promising and deserves to be implemented in a general systematic way in diabetic patients at renal risk, it may hardly represent the ultimate strategy to effectively prevent this devastating micro-vascular complication. From what we know of the complex, multifactorial mechanisms underlying the pathogenesis of diabetic renal disease, it is conceivable that, in order to obtain additional reno-protection, mechanisms that differ from glomerular hemodynamics should also be targeted. In fact, it is feasible that by combining SGLT2-is and RAAS-is, intra-glomerular pressure (and therefore, filtration fraction) may be reduced to a point beyond which further reduction may entail harm (i.e., GFR reduction) rather than a long-term benefit. In this perspective, other attractive agents such as neprilysin-inhibitors or GLP1R-A, which exert their renal protective action by means of different mechanisms, may soon prove their incremental value in providing kidney protection. Neprilysin inhibitors have potential in improving glucose homeostasis in patients with diabetes through increasing muscle sensitivity to glucose uptake and hepatic sensitivity to insulin. The simultaneous use of RAAS-is can make-up for the negative effects of increasing angiotensin II levels through neprilysin inhibition. The association of these two drugs could provide an efficacious cardiovascular and renal protection in combination with positive effects on glucose homeostasis [31].

## 4. Conclusions

In conclusion, optimal BP reduction by antihypertensive treatment currently remains the most effective and safe intervention to convey renal and cardiovascular protection in patients with diabetic kidney disease. RAAS-I should be considered the treatment of choice, although ideal BP values remain a matter of debate. Lower targets are recommended in the presence of overt albuminuria, a condition that entails greater risk of renal progression. A paradoxical J-curve relationship between BP reduction and renal morbidity may limit the benefit of aggressive treatment strategies, especially in the frail, elderly patients with non-albuminuric renal impairment.

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
