# Peer review of "Antihypertensive Treatment in Diabetic Kidney Disease: The Need for a Patient-Centered Approach"

_medicina, 2019, doi:10.3390/medicina55070382_

Round 1
Reviewer 1 Report
The work presented for the review concerns an interesting subject, although in some places this topic was, in my opinion, treated fairly generally. I mean, for example, SGLT2 inhibitors (here I suggest extending the article with the latest publications, e.g., Jaime A Davidson (2019)). Also, it would be interesting to add a few sentences about the simultaneous inhibition of neprilysin and modulation of the renin-angiotensin system to prevent diabetic nephropathy. On the other hand, in the case of albuminuria, I would add the sentence that in diabetic kidney disease (DKD) increased activation of the renin-angiotensin-aldosterone system (RAAS) contributes to renal fibrosis, article of Koszegi. I suggest accepting the article after minor changes.
I think that such additional information will enrich the article even more, and it will meet the high demands placed on "Medicina."
Author Response
We thank the reviewer for the constructive criticisms. We provided to extend the topic about SGLT2 (line 138) and neprilysin inhibitors (line 160), including the suggested paper (Davidson JA. Postgrad Med. 2019) We have emphasized how RAAS activity contributes to renal fibrosis as suggested by Koszegi (line 53).
Reviewer 2 Report
Well written paper on a clinically relevant topic. The sentence in Line 103 is not composed well and will need to be edited or reformatted.
A surprising omission would be what to do with the problem of diabetics with autonomic neuropathy and postural hypotension. This is a significant problem especially among the elderly. I acknowledge that not many good studies are done in this population. But the authors can share their approach too.
Author Response
We reformatted line 103 as requested by the reviewer. We developed the topic about the relevant problem of autonomic neuropathy and orthostatic hypotension in diabetic population (line 109) following reviewer’ suggestion.
This manuscript is a resubmission of an earlier submission. The following is a list of the peer review reports and author responses from that submission.